# Real-Time Defect Identification in Automotive Brake Calipers Using PCA-Optimized Feature Extraction and Machine Learning

**DOI:** 10.3390/s25216753

**Published:** 2025-11-04

**Authors:** Juwon Lee, Ukyong Woo, Myung-Hun Lee, Jin-Young Kim, Hajin Choi, Taekeun Oh

**Affiliations:** 1School of Architecture, Soongsil University, Seoul 06978, Republic of Korea; juwon0529@soongsil.ac.kr (J.L.); ukdragon@soongsil.ac.kr (U.W.); hun2m@soongsil.ac.kr (M.-H.L.); 2School of Civil, Environmental and Architectural Engineering, Korea University, Seoul 02841, Republic of Korea; 3Department of Physics, Incheon National University, Incheon 22012, Republic of Korea; jykim14@inu.ac.kr; 4Department of Safety Engineering, Incheon National University, Incheon 22012, Republic of Korea

**Keywords:** automotive brake caliper, machine learning, principal component analysis, NDT, defect identification, feature

## Abstract

This study aims to develop a non-contact automated impact-acoustic measurement system (AIAMS) for real-time detection of manufacturing defects in automotive brake calipers, a key component of the Electric Parking Brake (EPB) system. Calipers hold brake pads in contact with discs, and defects caused by repeated loads and friction can lead to reduced braking performance and abnormal vibration and noise. To address this issue, an automated impact hammer and a microphone-based measurement system were designed and implemented. Feature extraction was performed using Fast Fourier Transform (FFT) and Principal Component Analysis (PCA), followed by defect classification through machine learning algorithms including Support Vector Machine (SVM), k-Nearest Neighbor (KNN), and Decision Tree (DT). Experiments were conducted on five normal and six defective caliper specimens, each subjected to 200 repeated measurements, yielding a total of 2200 datasets. Twelve statistical and spectral features were extracted, and PCA revealed that Shannon Entropy (SE) was the most discriminative feature. Based on SE-centric feature combinations, the SVM, KNN, and DT models achieved classification accuracies of at least 99.2%/97.5%, 98.8%/98.0%, and 99.2%/96.5% for normal and defective specimens, respectively. Furthermore, GUI-based software (version 1.0.0) was implemented to enable real-time defect identification and visualization. Field tests also demonstrated an average defect classification accuracy of over 95%, demonstrating its applicability as a real-time quality control system.

## 1. Introduction

Automotive brake systems are a key component in ensuring vehicle stability and passenger safety. Among these, the caliper is a critical component that generates friction by pressing the brake pad against the disc. Because calipers are constantly exposed to repetitive mechanical loads, high temperatures, vibration, and shock. They are highly susceptible to microscopic structural damage or defects. These defects can lead to reduced braking power, abnormal noise, vibration, and uneven wear, posing serious safety threats. Therefore, advanced automated technologies capable of early detection and real-time diagnosis of defects in key components, such as calipers in Electric Parking Brakes (EPB), are crucial for securing quality competitiveness in the automotive industry.

Existing defect diagnosis studies have generally involved attaching additional sensors to components or performing manual measurements outside the production line, with their applicability mostly limited to surface-level defect detection [1]. Calipers are structural components that include complex internal mechanisms such as motors and pistons, making it difficult to accurately identify the presence of micro-crack based solely on external appearance. In addition, the limited availability of defect databases results in time-consuming and costly cause analysis. Therefore, reliable detection of such micro-defects during manufacturing requires non-contact and automated measurement techniques. To address these limitations, this study developed a non-contact Automated Impact Acoustic Measurement System (AIAMS) that integrates an automated impact hammer and a microphone. The proposed system enables stable and repeatable acquisition of impact-acoustic responses even under mass-production conditions, making it suitable for real-time defect detection in automotive brake calipers [2,3,4].

Meanwhile, recent studies have shown a rapidly growing trend toward approaches that combine sensor-based measurement with machine learning and deep learning techniques. This approach automatically extracts features from measurement signals and classifies the presence and type of defects through learning. Such methods are becoming essential for implementing smart factories and achieving high levels of quality control [5]. In signal processing, the Fast Fourier Transform (FFT) is widely used to convert time-domain signals to the frequency domain and quantify key amplitude and frequency components. Principal Component Analysis (PCA) is also widely used to reduce data dimensionality and maximize variance between normal and defective specimens [6].

In addition, studies have also introduced features such as wavelet transform-based multiresolution analysis and Shannon Entropy (SE) to sensitively detect abnormal patterns in noise-free signals, enhancing machine learning model learning efficiency and defect detection reliability. In terms of machine learning and deep learning models, traditional machine learning algorithms such as Support Vector Machine (SVM), K-Nearest Neighbor (KNN), Decision Tree (DT), and ensemble techniques, and deep learning models such as CNN (Convolutional Neural Network) and FCN (Fully Convolutional Neural Network) are being utilized. For example, Gnanasekaran et al. extracted histogram and wavelet features from brake pad vibration data and applied a nested dichotomy classifier to achieve classification accuracy of over 99% [7], and a study on EMB system emphasized that sensor sensitivity, environmental resistance, redundant design, and response speed are important factors in designing a fault diagnosis system [8]. Morales Matamoros et al. reviewed AI-based fault detection research across the manufacturing industry and analyzed that hybrid models such as PCA and random forests showed high practicality and accuracy [9], and Ding et al. presented a high detection rate and real-time processing possibility by applying PCA and Improved Reconstruction-Based Contribution Plots (IRBCP) techniques to the Brake Actuation Unit [10].

Recent research has focused on the need for precision on-site diagnostics by addressing diverse defect types beyond simple detection classification. Furthermore, field applicability for real-time, 100% inspection has improved through GUI-based real-time judgment interfaces that provide results within an average of 0.05–0.1 s [9]. This integrated approach directly contributes to improved productivity, reduced defect rates, and automated quality management in smart factory environments.

In line with this objective, an integrated real-time defect diagnosis approach is proposed to overcome issues such as insufficient automation and inconsistent inspection quality in mass-production environments. The primary objective of this study is to acquire impact response data from actual automotive brake calipers-one of the most safety-critical components-and to enhance real-time diagnostic performance and applicability to production lines through FFT-PCA-based feature extraction and comparative evaluation of multiple machine learning classifiers to identify the optimal defect classification model.

## 2. Related Work

Research on automotive brake system and caliper fault diagnosis has evolved from the beginning, primarily centered on vibration and acoustic signal analysis. Baydar and Ball [1] demonstrated that acoustic signals could sensitively reflect fault characteristics in specific high-frequency ranges by comparing and analyzing vibration and acoustic signals in the time-frequency domain using the Wigner-Ville Distribution. Singh and Parey [5] showed that the limitations of traditional analysis techniques could be overcome by precisely characterizing abnormal signals using wavelet transform coefficients and achieving high classification performance through Artificial Neural Networks (ANNs).

Machine learning-based diagnostic techniques subsequently became mainstream, significantly improving diagnostic performance. The approach of quantifying signal features using FFT and PCA, reducing complexity through dimensionality reduction, and then applying classifiers such as SVM, KNN, DT, and Random Forest is widely utilized [6,9]. Furthermore, Kadlec et al. [11] demonstrated the feasibility of real-time diagnosis by applying FFT-PCA-SVM to process signals. Gnanasekaran et al. [7] emphasized the potential of machine learning-based diagnosis by representing brake pad fault data using histogram and wavelet methods and achieving an accuracy of over 99% with a Nested Dichotomy classifier. Morales Matamoros et al. [9] analyzed that PCA, Random Forest, and Ensemble learning can stably handle complex data in a manufacturing environment, and Ding et al. [10] achieved nearly perfect real-time detection of faults in a Brake Actuation Unit by applying PCA and IRBCP (Integrated Residual-Based Contribution Plot).

Deep learning-based approaches are also expanding in this field. Methods based on CNN and FCN can detect fault location and type in an end-to-end manner by transforming time-series data into an image format or inputting it directly [12]. For instance, Wang et al. accurately predicted brake pad wear status with an R^2^ of 0.9944 and an RMSE of 0.0023 using a CNN-LSTM-GTO model [13]. Viswanathan et al. [14] detected brake anomalies with 98% accuracy by combining ARMA (Autoregressive Moving Average) and an ensemble method, and Das [15] proposed a system that automates acoustic-based fault classification. Jensen [16] developed an RUL (Remaining Useful Life) model that uses braking force data to predict brake pad wear and remaining service life. Digital twin-based diagnostic techniques are also emerging. Gaurav [17] proposed a framework that links sensor data with MATLAB (version 24.2.0.2712019) simulation to detect anomalies in the braking system. A comprehensive review summarized that brake condition diagnosis, combining vibration and noise-based data with machine learning techniques, is crucial for improving vehicle safety [18,19].

However, existing studies have primarily focused on results derived from limited laboratory-scale data, and most provide insufficient discussion of processing speed, a key factor for real-time defect identification in mass production environments (Table 1). Comprehensive solutions integrating complex fault classification, low-latency real-time processing, user-friendly GUI, and continual learning, which are required in a real production environment, have not been sufficiently researched [20]. Discrepancies exist between current research and field reality concerning issues like data imbalance, model generalization ability, and real-time feedback integration. This study addressed these gaps by:Data collection: Collecting total 2200 impact response datasets from normal, internal fault specimens of actual caliper components to establish a learning and validation environment based on practical field data.Feature engineering and modeling: Utilizing FFT-PCA-based feature extraction and dimensionality reduction, followed by a comparative evaluation of machine learning classifiers such as SVM, KNN, and DT to derive the optimal combination.Real-time solution development: Developing GUI-based real-time software (version 1.0.0) capable of outputting results within an average of 0.05–0.1 s and providing visual feedback for fault detection.Validation and proof of concept: Achieving a discrimination accuracy of 97% and a processing time of 117.6 ms through on-site testing witnessed by a certification body, thereby demonstrating its applicability to production line full inspection.

Through these efforts, this research proposes an integrated real-time caliper fault diagnosis system, which was lacking in prior literature, and shows its potential as a foundational technology for implementing smart factories and automating quality control, with extensibility to other components like EPB, discs, and motors.

## 3. Methodology

### 3.1. Suggested System

The primary objective of this study is to propose the development of an integrated real-time caliper diagnostic system. The overall architecture of the proposed system is shown in Figure 1, and it consists of the following nine steps.
Specimen conveyance: In this step, the caliper specimen is automatically conveyed to the measurement position using a conveyor mechanism integrated into the AIAMS. The conveyor operates at a constant speed to maintain consistent alignment and positioning of each specimen while minimizing manual intervention. This enables continuous and uniform measurements under conditions similar to those in an actual production environment.Auto-impact based data collection: When the specimen reaches the designated position, an automated impact hammer delivers mechanical impact with a constant force and interval. The hammer is programmed to strike the same location repeatedly to ensure measurement repeatability. This step provides reliable impact-acoustic response signals by exciting the caliper structure under consistent conditions.Non-contact measurement: The impact-acoustic response signals of the caliper are measured by a microphone positioned at a fixed distance using a non-contact method. This step minimizes external interference and ensures data stability and repeatability. The acquired acoustic signals contain resonant characteristics and structural information of the caliper.Fast Fourier Transform: The recorded time-domain signal is converted into the frequency domain using the Fast Fourier Transform (FFT). This transformation allows extraction of the resonant frequencies and spectral components that reflect the structural integrity and dynamic behavior of the caliper. The frequency-domain data obtained here serve as the foundation for subsequent feature extraction.Frequency band selection: The frequency spectra of normal and faulty specimens are cumulatively analyzed to identify regions exhibiting distinct amplitude differences between the two groups. As a result of this comparative analysis, the 4.5–5.0 kHz range was determined to be the optimal frequency band for discriminating between normal and defective specimens. This selected frequency band is used as the key analysis region for fault identification.Feature Extraction: Within the selected frequency range, twelve statistical and spectral features are extracted, including MAX, RMS, CF, STD, K, SE, FM0, M6A, M8A, CLFIPI, and MF. Each feature comprehensively represents the energy distribution, statistical characteristics, and dynamic behavior of the impact-acoustic signal, serving as a critical variable for fault discrimination.PCA-based feature selection: PCA is performed on the extracted twelve features to identify the variables that most effectively explain data variability. As a result, SE exhibited the highest contribution to the first principal component (PC1), demonstrating strong discriminative capability. Accordingly, SE was selected as the key feature and combined with the remaining eleven features to construct a total of eleven feature combinations.Machine learning: Based on the derived feature combinations, three machine learning models that included SVM, KNN, and DT were trained and validated. The dataset was divided into an 8:2 ratio for training and validation to evaluate model performance. The results confirmed that the proposed approach achieved high accuracy and stability in classifying normal and defective specimens.GUI development: Finally, a Graphical User Interface (GUI) was developed for real-time defect identification and visualization. The GUI integrates the trained models, automatically classifying the caliper condition immediately after measurement and displaying the results. This enables users to intuitively monitor defect status, supporting the realization of a fully automated inspection system in industrial environments.

### 3.2. Automated Impact-Acoustic Measurement System

In this study, AIAMS was developed to automate the integrity inspection of manufacturing faults in automobile components. The AIAMS is a non-contact measurement system that combines an automated impact hammer (330AE-05, AI Systems, Gyeonggi-do, Republic of Korea) and a microphone (378C01, PCB Piezotronics, Inc., Depew, NY, USA). The automated impact hammer strikes the caliper with a constant force and period, and the microphone non-contactly measures the resulting acoustic signal. Furthermore, the caliper is placed on a conveyor belt that rotates clockwise at a constant speed, ensuring the impact point remains consistent. This configuration minimizes external interference and allows for the consistent collection of data, regardless of the caliper’s surface condition or position changes. In particular, a detailed schematic of the AIAMS is presented in Figure 2, and the key specifications of its main components such as the automated impact hammer and the microphone are summarized in Table 2.

### 3.3. Principal Component Analysis

PCA is one of the most widely used data analysis techniques in various fields utilizing machine learning and artificial intelligence [23]. PCA is a method for reducing the dimensionality of a dataset composed of numerous correlated variables while retaining as much of the data’s variability as possible. This process is accomplished by transforming the dataset into a new set of variables called Principal Components (PC) [21].

Principal Components are derived by computing the eigenvectors and eigenvalues of the covariance matrix and then arranging the eigenvectors in descending order according to the magnitude of their eigenvalues. These resulting principal components are uncorrelated, and the PC1 accounts for the largest possible variance in the original variables [22].

In this study, PCA was performed to identify the key features among the twelve extracted features. The twelve features obtained from each specimen were normalized using the Min-Max method to prevent bias caused by unit differences and to ensure balanced scaling among features. The normalized data were then concatenated to construct the input matrix for the PCA function. Eigenvalue decomposition of the covariance matrix was performed to obtain the principal component coefficients (coeff), principal component scores (score), and eigenvalues (latent). The explained variance ratio of each principal component was calculated to evaluate the proportion of total variance represented by each component, while the cumulative variance ratio was used to identify the minimum number of components required to capture the dominant data variability.

In addition, the absolute values of coeff were analyzed to identify the features that contributed most to each principal component. As a result, PC1 explained approximately 55% of the total variance, and feature SE exhibited the highest coeff value, confirming its dominant influence on data variability. Based on this PCA-driven quantitative comparison, SE was selected as the most discriminative feature for fault identification. The detailed results of this analysis are presented in Section 4.2.2.

### 3.4. Machine Learning

#### 3.4.1. Support Vector Machine

Machine learning is a multidisciplinary research area utilized in various academic fields, including cognitive science, computer science, statistics, and optimization. Machine learning is generally divided into supervised and unsupervised learning. Supervised learning is a method of training and building a model based on a labeled dataset that includes class information, whereas unsupervised learning is a method of exploring latent structures or patterns within an unlabeled dataset.

A key advantage of machine learning is its ability to effectively detect complex patterns even in large-scale datasets, enabling fault discrimination even in noisy data environments. This characteristic is particularly advantageous for discerning minute faults, such as those found in a brake caliper. In this study, SVM, KNN, and DT models were applied to classify normal and faulty calipers based on feature values extracted from the impact-acoustic data.

Among these, SVM is one of the most representative supervised learning-based classification techniques, possessing excellent generalization performance, the ability to derive optimal boundaries, and strong discriminative power. In practice, SVM has been applied across various fields, including handwriting recognition, credit card fraud detection, speaker identification, and facial recognition. In this study, SVM was used to learn the optimal hyperplane between the normal and faulty calipers, thereby maximizing the separation between classes. The data points closest to the hyperplane are called support vectors, and the goal of training is to maximize the margin between these vectors and the hyperplane. The margin is directly related to the weight vector w and is defined as 1/∣∣w∣∣. Therefore, by securing this margin, the optimal classifier minimizes classification errors and can stably discriminate faults even under various noisy conditions that may occur on an actual production line [24,25].

#### 3.4.2. K Nearest Neighbor

KNN is one of the simplest and most intuitive machine learning algorithms, used in various fields such as density estimation and clustering. In this study, for each impact-acoustic data sample, the k nearest neighbors were identified, and the sample was classified as normal or faulty based on the majority vote principle.

The greatest advantage of KNN is that it is a non-parametric model, meaning it does not require any prior assumptions about the distribution of data. This is a significant benefit, as caliper faults encountered in a manufacturing environment do not necessarily follow a specific distribution. Furthermore, its simple training process and low computational load make it highly applicable in a real-time quality inspection environment. Although the fault data in this study was difficult to distinguish due to its similarity with normal signals, KNN was able to more stably discriminate fault signals located at the boundary values by reflecting the collective characteristics of its surrounding neighbors [26,27].

#### 3.4.3. Decision Tree

The DT is a model that offers high interpretability and a judgment structure similar to human reasoning through a simple rule-based classification framework. DT works by recursively partitioning data based on continuous or categorical attributes to generate a series of decision rules that distinguish between normal and faulty conditions.

In practical industrial applications, such as caliper fault diagnosis, model reliability and explainability are crucial. DTs have the advantage of having decision rules that are intuitively understandable compared to black-box models like neural networks, making it easy for inspectors or field engineers to interpret and trust the results. Furthermore, DTs exhibit relative robustness against noise in the training data and were able to present clear classification rules (e.g., entropy values in a specific frequency range) even for the diagnosis of subtle caliper faults [28,29].

## 4. Experiment Validation

### 4.1. Experimental Configurations and Specimen

To verify the feasibility of the proposed method, the brake caliper was mounted onto the AIAMS, and measurements were conducted. Figure 3 shows the configuration diagram of the AIAMS, which was designed for evaluating the integrity of the caliper. This system maximized the practicality and field applicability of the measurement data by adopting the conveyor belt, which is widely used in actual manufacturing sites. The caliper was designed to move along the conveyor belt at a constant speed while maintaining a fixed interval, preventing signal interference between consecutively measured specimens.

As shown in Figure 4, the specimens used for verification consisted of 5 normal calipers and 6 faulty calipers produced on an actual production line, which increased the practical relevance of this study. Notably, all faulty specimens contained micro-defects, which, as confirmed in Figure 5, were so small that they were virtually impossible to identify visually in a mass-production environment.

Each caliper was measured 200 times repeatedly at approximately 5 s intervals, resulting in the establishment of 2200 impact response datasets. This repeated measurement design not only ensured the statistical reliability of the measured data but also empirically demonstrated that the proposed AIAMS can be applied for the early detection of micro-defects in a real production environment.

### 4.2. Experimental Results

#### 4.2.1. Preprocessing

Figure 6 presents a representative example from a total of 2200 datasets acquired from the caliper specimens, showing both the time-domain signal and the frequency spectrum obtained through the Fourier Transform. Specifically, Figure 6a,c corresponds to a normal specimen, while Figure 6b,d represents a faulty specimen. A comparison of the time-domain signals between the normal and faulty specimens revealed that it was difficult to confirm any distinct differences by visual inspection alone. Furthermore, even the frequency spectra showed only minor differences, which were insufficient for reliably determining the presence of a fault. This suggests that there are limitations of existing resonant frequency-based analysis for classifying micro-defects.

To overcome these limitations, this study, as shown in Figure 7, cumulatively aggregated the amplitude information from the frequency spectra of the normal specimens and the faulty specimens independently. By collectively summing the spectra of the normal specimens and treating the faulty specimens’ spectra identically, the dynamic characteristics between the two groups could be more clearly compared. As a result, the most pronounced amplitude difference between the normal and faulty specimens was observed in the 4.5–5.0 kHz frequency band.

Therefore, this study sets this specific frequency band as the primary analysis band for fault discrimination. Subsequent feature extraction and classification model training were performed using this frequency domain as the core input variable, which allowed for the maximization of micro-defect detection accuracy in the caliper.

#### 4.2.2. Feature Extraction

In this study, the impact response signal acquired from the caliper specimens was analyzed in the frequency domain to derive a total of 12 features that reflect various statistical and spectral characteristics. The definitions of these features are presented in Equations (1)–(12), where x represents the impact response signal, xi is the i-th sample of the signal, and N is the total number of samples [30].

Since a single feature only explains a specific aspect of the signal, the information it can provide for fault detection is limited. For example, while MAX (Maximum) or RMS (Root Mean Square) can effectively represent the overall signal energy, they are insufficient for capturing high-order statistical properties closely associated with faults, such as asymmetry, kurtosis, or the irregularity of the distribution. Accordingly, this study aimed to more clearly identify the dynamic behavioral differences between normal and faulty specimens by comprehensively applying multiple features that reflect various aspects of the signal.

Based on the total datasets of 2200, 12 features were calculated, yielding a total of 26,400 feature values. These values were separated into normal and faulty groups, and their distributional characteristics were visualized using a boxplot (Figure 8). The analysis confirmed a distinct difference between the two groups for most features. In particular, SE showed a significant tendency to increase in faulty specimens compared to normal specimens. This means that SE can effectively capture the irregularity and variability of the signal energy distribution, suggesting it can serve as a more sensitive and discriminative feature for fault detection than other features.(1)MAX(Maximum value)=max(x)(2)$$RMS(Root\;mean\;square)=\sqrt{\frac{1}{N}[\sum_{i=1}^{N}{\left(x_{i}\right)}^{2}]}$$(3)$$CF(Crest\;factor)=\frac{MAX}{RMS}$$(4)$$STD(Standard\;deviation)=\sqrt{\frac{1}{N-1}[\sum_{i=1}^{N}{\left(x_{i}-\bar{x}\right)}^{2}]}$$(5)$$K(Kurtosis)=\frac{N\sum_{i=1}^{N}{\left(x_{i}-\bar{x}\right)}^{4}}{{(\sum_{i=1}^{N}{\left(x_{i}-\bar{x}\right)}^{2})}^{2}}$$(6)$$FM0=\frac{\max\left(x\right)-min(x)}{sum(x)}$$(7)$$SE(Shannon\;entropy)=-\sum_{j}^{J}p_{j}logp_{j}$$(8)$$M6A=\frac{N^{2}\sum_{i=1}^{N}{\left(x_{i}-\bar{x}\right)}^{6}}{{(\sum_{i=1}^{N}{\left(x_{i}-\bar{x}\right)}^{2})}^{3}}$$(9)$$M8A=\frac{N^{2}\sum_{i=1}^{N}{\left(x_{i}-\bar{x}\right)}^{8}}{{(\sum_{i=1}^{N}{\left(x_{i}-\bar{x}\right)}^{2})}^{4}}$$(10)$$CLF(Clearance\;factor)=\frac{MAX}{(\frac{1}{N}{\sum_{i=1}^{N}\sqrt{\left|x_{i}\right|})}^{2}}$$(11)$$IPI(Impulse\;indicator)=\frac{MAX}{\frac{1}{N}\sum_{i=1}^{N}\left|x_{i}\right|}$$(12)$$MF(Mean\;frequency)=\frac{1}{N}\sum_{i=1}^{N}x_{i}$$

#### 4.2.3. Feature Selection and Combination

The objective of the PCA performed in this study was to identify the core features that could most effectively distinguish between the normal and faulty groups by deriving the principal components that best explain the data variability out of the total 26,400 feature values. Figure 9a shows the individual explained variance and cumulative explained variance ratios explained by each principal component, demonstrating that the majority of the data’s variability can be accounted for by only a small number of principal components. Specifically, PC1 was confirmed to act as the most important information axis, explaining approximately 55% of the total data variability.

Furthermore, Figure 9b presents a heatmap showing the relative contribution of each feature to PC1 through PC5, based on the results from Figure 9a. The analysis reconfirmed that PC1 predominantly explains the data variability. SE emerged as the feature with the largest contribution to PC1. This signifies that SE effectively captures the irregularity and energy distribution difference between normal and faulty specimens, suggesting its critical role as a core feature in fault classification.

Therefore, this study selected SE as the primary feature. This feature was then combined with each of the remaining 11 features to create a total of 11 feature combinations (FCs) (Table 3). Subsequently, each FC was utilized as the training and validation data for the SVM, KNN, and DT classification models. The dataset, consisting of 2200 impact response measurements acquired from 11 caliper specimens (5 normal and 6 defective) through 200 repeated tests at 5 s intervals, was divided into an 8:2 ratio for training and validation, respectively, to evaluate model performance.

#### 4.2.4. Training and Prediction

In this study, the classification performance of the SVM, KNN, and DT models was evaluated based on the derived feature combinations (FCs). Figure 10, Figure 11 and Figure 12 present representative results for classification using FC3 (SE-CF). The graph on the left shows the decision boundary and data distribution for each machine learning model, confirming a clear separation between the normal and faulty data groups. Notably, the red squares indicate the accurately classified data, demonstrating that most of the training and validation samples were correctly classified.

The graph on the right presents the classification accuracy for the normal and faulty groups through the confusion matrix. The analysis revealed that the SVM model achieved 100% accuracy for both normal and faulty specimens. The KNN model recorded an accuracy of 99.5% for normal specimens and 100% for faulty specimens, while the DT model recorded 100% accuracy for normal specimens and 99.6% for faulty specimens.

Furthermore, an analysis of the minimum classification accuracy across all feature combinations (FC1-FC11) revealed that SVM secured at least 99.2% for the normal group and 97.5% for the faulty group. KNN achieved at least 98.8% for the normal group and 98.0% for the faulty group, and DT achieved at least 99.2% for the normal group and 96.5% for the faulty group. These results prove that the SE-based feature combinations effectively reflect the characteristics of both normal and faulty data and can achieve excellent classification performance across all three machine learning models.

The detailed classification accuracies for each feature combination are summarized in Table 4, allowing for a comprehensive comparison of model performance.

#### 4.2.5. Ablation Study

To further verify the robustness and design rationale of the proposed defect classification system, two additional ablation studies were conducted, and the results are summarized in Table 5 and Table 6.

In the first ablation study, the entire frequency range (0–12.8 kHz) was analyzed instead of the primary analysis band (4.5–5.0 kHz) defined in Step frequency band selection of the proposed method. The overall classification accuracy decreased when the full range was used, which can be interpreted as the inclusion of redundant and noisy components that reduced the discriminative performance of the models. As shown in Table 3, the SVM model achieved average accuracies of 74.8% and 64.1% for the normal and defective specimens, respectively. The KNN model recorded 74.3% and 66.8%, while the DT model achieved 73.9% and 66.6%. These results confirm that the frequency band determined in the Frequency band selection step of the proposed system is optimized for defect detection and most effectively represents the differences between normal and defective specimens.

In the second ablation study, feature normalization was applied prior to machine learning. While the baseline method used unnormalized features to preserve the intrinsic magnitude differences between normal and defective specimens, in this ablation study, the normal and defective datasets were independently normalized using the Min-Max method. Consequently, the original feature contrast between classes was reduced, resulting in lower overall classification accuracy. As presented in Table 4, the SVM model achieved average accuracies of 96.6% (normal) and 97.6% (defect), the KNN model recorded 96.2% and 97.8%, and the DT model obtained 95.8% and 97.6%, respectively. These findings indicate that the amplitude difference between normal and defective signals provides essential discriminative information, and normalization tends to obscure this difference, thereby weakening model separability.

Therefore, these ablation studies demonstrate that the selected frequency band and the use of unnormalized features are key design factors that ensure the high accuracy and stability of the proposed defect classification system.

## 5. Discussion

### 5.1. Limitations

The proposed AIAMS was developed to detect micro-defects in automotive brake calipers through automated impact-acoustic measurements during conveyor transfer, achieving real-time classification within approximately 117.6 ms by integrating PCA-based feature selection and machine learning techniques. The system enables objective and quantitative evaluation through AI-based analysis, and the learning model can be regarded as multi-modal, since it employs combinations of multiple features centered on SE rather than a single feature. This configuration allows for stable signal acquisition and high-precision defect discrimination even under mass-production conditions, ensuring strong robustness and generalization performance across varying operational environments. However, several considerations and subsequent research are necessary for its application in real industrial settings.

First, diversification of defect types is necessary. This study primarily focused on micro-defects to verify the feasibility of the proposed approach. In this context, the system architecture was designed to be applicable to various casting-related defects such as short fill, cold shut, surface crack, and shrinkage cavity (Figure 13). Future research will expand the experimental scope to include these additional defect types to comprehensively evaluate the applicability and scalability of the proposed method. The analytical framework is expected to be further expanded through the incorporation of additional defect categories into model training, enabling continuous evolution and adaptation to a broader spectrum of defect characteristics. The framework is also planned to be applied to other critical automotive components such as shafts, gears, and bearings to evaluate its generalization performance for components with different geometries and material properties. In addition, the proposed system is intended to be developed into an end-to-end defect recognition framework with enhanced real-time processing capability through the integration of deep learning models such as CNN-LSTM.

Second, several practical factors must be considered for industrial application. Although factory environments may exhibit higher levels of noise, vibration, and temperature fluctuations, these factors are mostly concentrated in the low-frequency range (below several hundred Hz). Since the proposed system operates in the high-frequency (kHz) domain and measures impact responses near the caliper, such low-frequency disturbances can be effectively suppressed using simple signal-processing techniques such as high-pass filtering. To ensure long-term operational stability, reproducibility of impact force, precision in sensor alignment, and a regular on-site calibration routine must be maintained.

In actual industrial environments, the structural geometry of brake calipers is largely standardized across manufacturers to meet performance and safety requirements. Furthermore, calipers produced by the same manufacturer are typically cast using identical molds and homogeneous alloy compositions, and they undergo consistently controlled surface treatment processes. As a result, the variation in acoustic response characteristics among calipers of the same type is minimal, and the influence of shape or material differences on the measured signals is expected to be negligible. Considering these factors, the proposed method is expected to be stably applicable to calipers with diverse shapes and material properties used in real production environments. Nevertheless, future studies should comprehensively consider these factors through additional experiments and validation.

In real manufacturing environments, the geometry and material composition of calipers show minimal variation across manufacturers. Calipers from the same manufacturer are typically cast using identical molds with homogeneous alloy compositions and undergo nearly flawless surface treatment processes, resulting in highly uniform products. Under such standardized and controlled conditions, the proposed system can be reliably applied to detect rare but critical defects that may still occur during large-scale production. Although the occurrence of defects in caliper manufacturing is maintained at an extremely low level through rigorous quality control, the safety-critical nature of this component necessitates precise and objective detection of even minute faults. Therefore, AIAMS satisfies industrial demands for high reliability and accuracy in automated, non-contact inspection, and its framework can be further enhanced through the continuous accumulation and learning of defect data, leading to improved robustness and scalability for broader application.

Finally, since this study was validated using metallic calipers, additional parameter calibration may be required when applying the system to non-metallic or composite components to compensate for differences in acoustic propagation characteristics.

### 5.2. Comparative Study

The feasibility and superiority of the proposed multi feature model were verified through a comparative study against a conventional single feature model in defect detection. The single feature model applies each feature independently to train the machine-learning classifiers and evaluates the resulting classification accuracy [31]. The analysis pipeline was aligned with that of the multi feature model, except that the PCA-based key-feature selection and the feature-combination step for constructing the multi feature were omitted. Accordingly, for comparison with the eleven feature combinations of the proposed method, additional classifications were performed by using each of the eleven individual features excluding SE as a single input to SVM, KNN, and DT.

In the analysis, the single feature model yielded mean classification accuracy and variance of 68.0%/203.5 with SVM, 68.6%/154.4 with KNN, and 68.5%/148.4 with DT, which are markedly inferior to the previously obtained results of the multi feature model (≈99% accuracy/0.73 variance). These results demonstrate that the proposed multi feature model, by integrating multiple features, substantially enhances model robustness and classification reliability. The detailed results are presented in Figure 14, where each black dot represents an individual accuracy observation, and the red horizontal line inside each box indicates the median value of the distribution.

## 6. Conclusions

In this study, AIAMS was developed for the detection of manufacturing defects in automotive brake calipers. Based on the experimental analysis, the following conclusions could be drawn.
(1)The proposed AIAMS integrates an automated impact hammer, a microphone, and a conveyor system to allow for the consistent and repetitive acquisition of impact-acoustic signals in an environment similar to a production site. This system was validated using a total of 11 caliper specimens, consisting of five normal specimens and six faulty specimens.(2)Each specimen underwent 200 repeated measurements, resulting in a total dataset of 2200 impact response datasets being secured. From this data, 12 statistical and spectral features were extracted in the frequency domain. The PCA results confirmed that SE was the highly discriminative feature that most effectively captured the irregular and energy distribution differences between the normal and faulty specimens.(3)Feature combinations (FC1-FC11) centered around SE were constructed to train and validate the SVM, KNN, and DT models. As a result, SVM achieved an accuracy of 99.2% or higher for normal specimens and 97.5% or higher for faulty specimens. KNN achieved 98.8% and 98.0% or higher, respectively, and DT achieved 99.2% and 96.5% or higher, respectively. This demonstrates that the proposed method consistently maintains stable and reliable classification performance regardless of the specific feature combination applied.(4)The research results validate that the proposed AIAMS can detect micro-defects in automotive brake calipers in a highly reliable and automated manner by combining PCA-based feature selection with machine learning techniques. Previous studies have mostly relied on laboratory-scale data, and discussions regarding real-time defect identification; particularly, the processing speed aspect required for applications in mass production processes have been relatively limited. In contrast, the proposed system integrates a conveyor-based non-contact impact-acoustic measurement structure capable of repeated and automated inspections. By applying the Shannon entropy-based high-precision feature extraction technique, the system achieved real-time defect detection with an average processing time within 117.6 ms, demonstrating its suitability for practical implementation in automotive manufacturing environments that demand both high accuracy and rapid inspection speed.

## Figures and Tables

**Figure 1 sensors-25-06753-f001:**
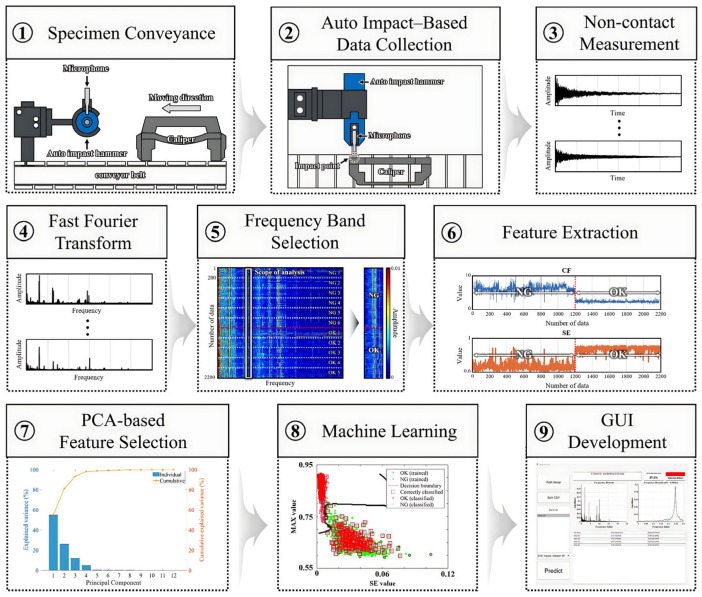
Flowchart of the overall analysis process of impact-acoustic data using AIAMS.

**Figure 2 sensors-25-06753-f002:**
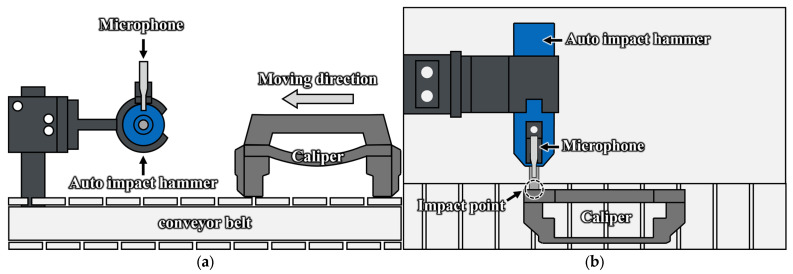
Schematic diagram of AIAMS: (**a**) Side view before impact; (**b**) Top view of the impact process.

**Figure 3 sensors-25-06753-f003:**
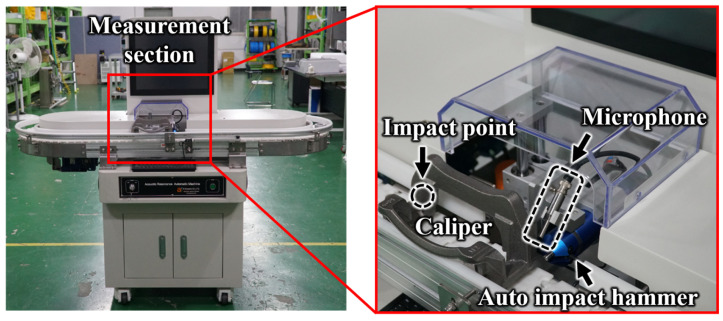
Experimental setup.

**Figure 4 sensors-25-06753-f004:**
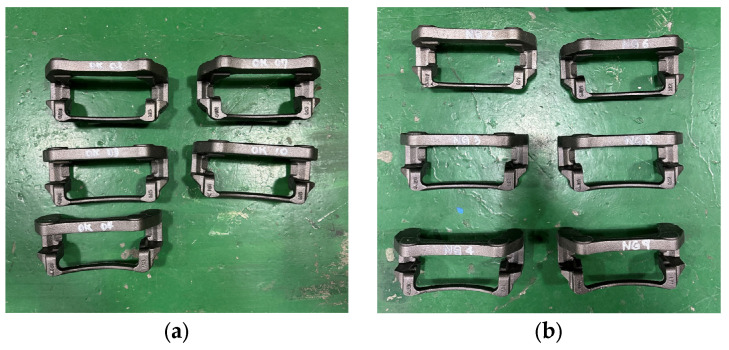
Type of brake caliper specimens: (**a**) Normal specimen; (**b**) Micro-defect specimen.

**Figure 5 sensors-25-06753-f005:**
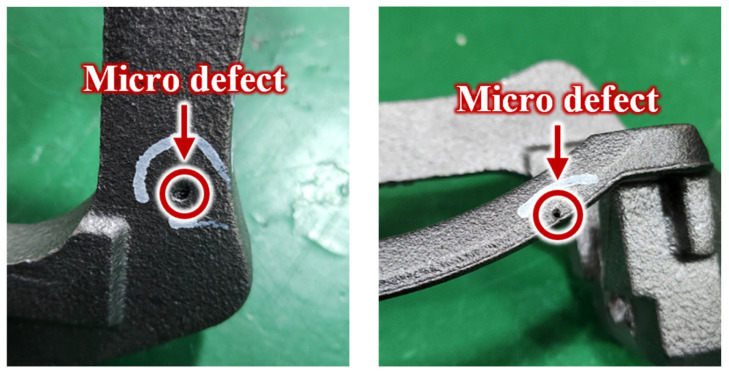
Photographic of micro-defects in brake calipers.

**Figure 6 sensors-25-06753-f006:**
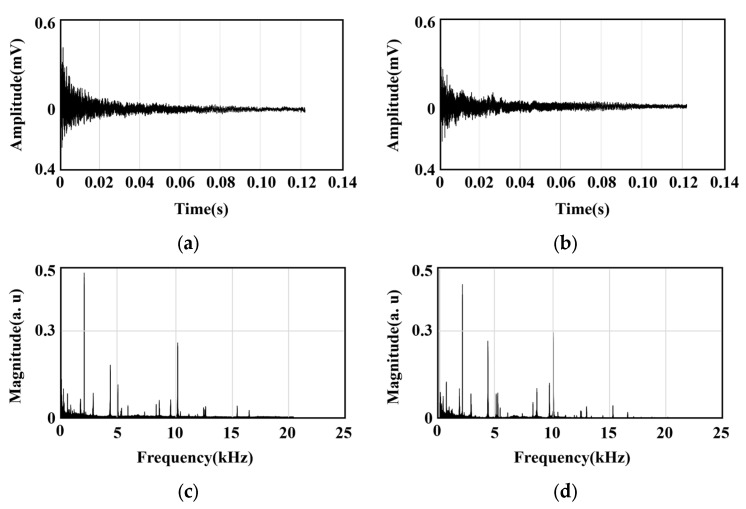
Measured signals by AIAMS: (**a**) time domain of normal specimen; (**b**) time domain of defect specimen; (**c**) frequency spectrum of normal specimen; (**d**) frequency spectrum of defect specimen.

**Figure 7 sensors-25-06753-f007:**
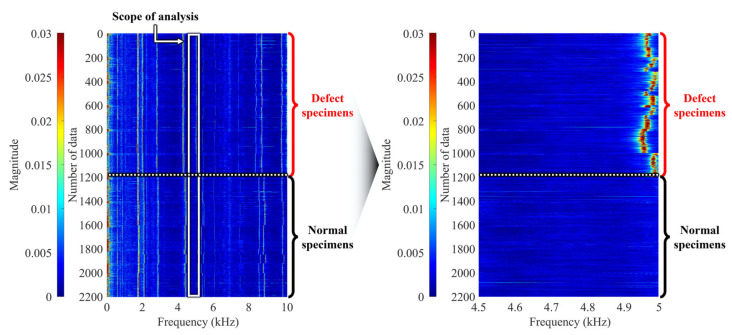
Accumulated frequency spectrum distributions of normal and defect specimens.

**Figure 8 sensors-25-06753-f008:**
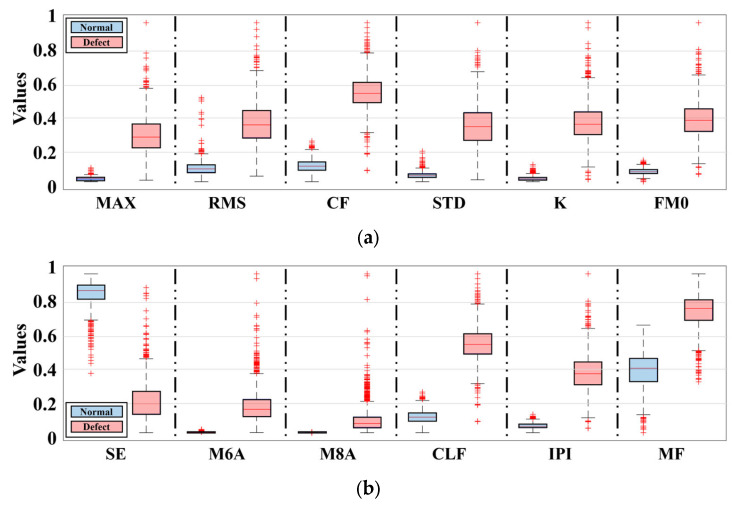
Feature extraction result: (**a**) Feature; MAX, RMS, CF, STD, K, FM0; (**b**) Feature; SE, M6A, M8A, CLF, IPI, MF.

**Figure 9 sensors-25-06753-f009:**
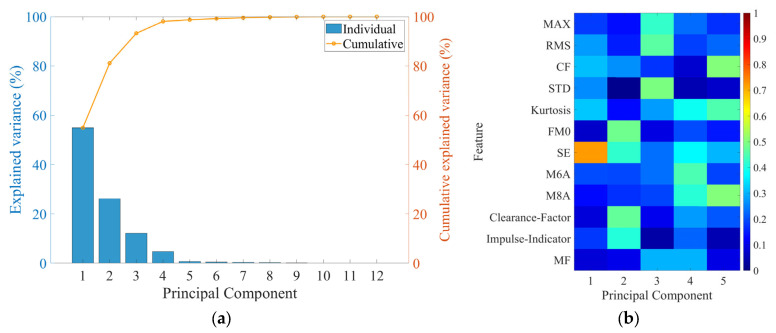
PCA result; (**a**) Explained variance ratio, (**b**) Feature loadings on principal components.

**Figure 10 sensors-25-06753-f010:**
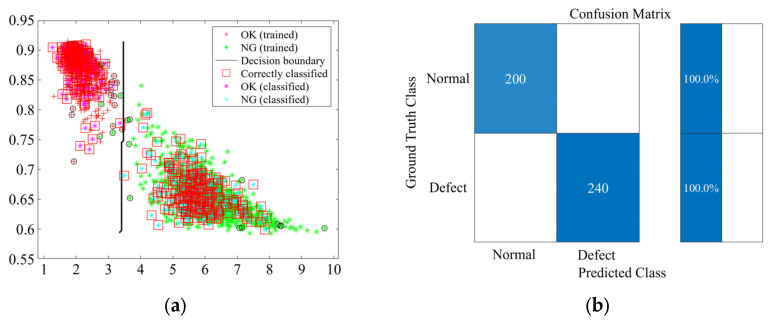
SVM training and prediction results for FC3 (SE-CF): (**a**) Classification result; (**b**) Confusion matrix.

**Figure 11 sensors-25-06753-f011:**
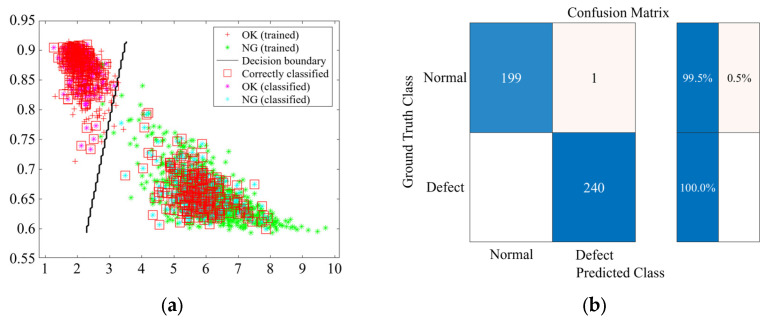
KNN training and prediction results for FC3 (SE-CF): (**a**) Classification result; (**b**) Confusion matrix.

**Figure 12 sensors-25-06753-f012:**
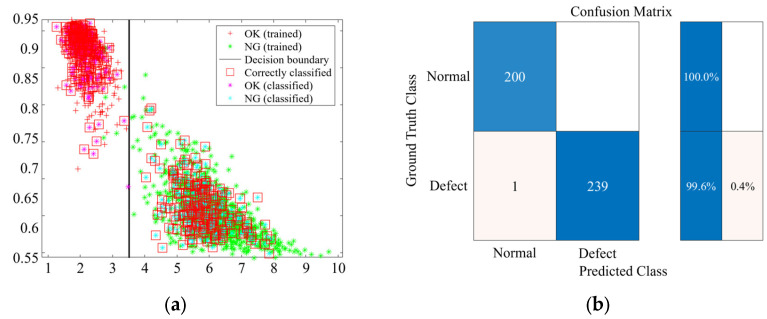
DT training and prediction results for FC3 (SE-CF): (**a**) Classification result; (**b**) Confusion matrix.

**Figure 13 sensors-25-06753-f013:**
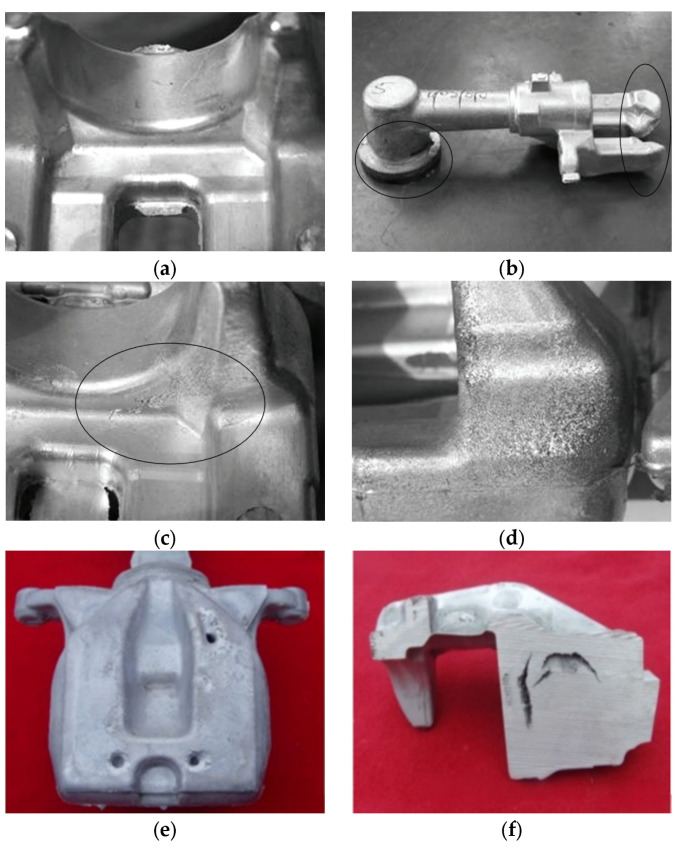
Example of brake caliper manufacturing defects (**a**) Normal; (**b**) Short fill; (**c**) Cold shut; (**d**) Surface finish; (**e**) Shrinkage cavity; (**f**) Shrinkage crack.

**Figure 14 sensors-25-06753-f014:**
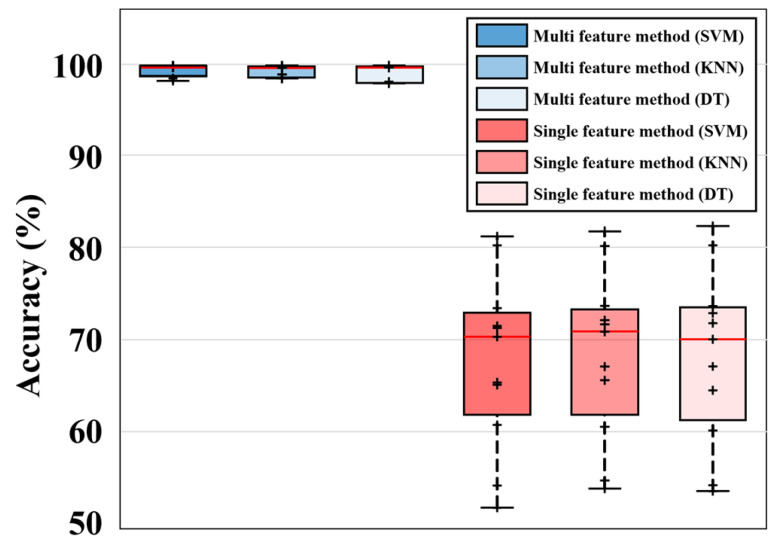
Comparison of classification accuracy between proposed multi feature model and single feature model.

**Table 1 sensors-25-06753-t001:** Comparison with previous studies.

Article	Data Type	Processing Speed	Accuracy	Remarks
[10]	Contact vibration signal of brake pad	Not specified	99.45%	CBND classifier with wavelet feature
[13]	Contact pressure signal of brake cylinder	Not specified	99.00%	IRBCP with Mutual Residual and PCA
[15]	Contact sensor signal of semiconductor factory equipment	Not specified	99.48%	MTS-CNN
[17]	Contact vibration signal of disc brake and drum brake	Not specified	98.00%	Ensemble voting classifier with ARMA feature
[18]	Non-contact acoustic signal of hydraulic brake system	Not specified	97.80%	Sequential model-based CNN
[21]	Contact vibration signal of hydraulic brake system	Not specified	98.36%	SVM classification with statistical feature
[22]	Vibration signal of piston rod and steering rack	Not specified	100.00%	PCA-SVM-BiLSTM
AIAMS	Non-contact impact-acoustic signal	117.6 ms	97.00%	Real-time defect detection machine learning model based on PCA and feature combination

**Table 2 sensors-25-06753-t002:** Specifications of auto-impact hammer and microphone.

Device	Parameter	Specification
Auto-impact hammer	Durability	1,000,000 cycles
	Stroke	5 mm, 15 mm
	Force range	up to 100 N
	Size	137 × 38 × 38 mm
Microphone	Sensitivity	10 mV/Pa
	Frequency range	20~16,000 Hz
	Excitation voltage	18 to 30 VDC
	Size	13.2 × 69.9 mm

**Table 3 sensors-25-06753-t003:** Feature combination list (11EA).

Feature Combination	Feature 1	Feature 2
FC1	SE	MAX
FC2	SE	RMS
FC3	SE	CF
FC4	SE	STD
FC5	SE	K
FC6	SE	FM0
FC7	SE	M6A
FC8	SE	M8A
FC9	SE	CLF
FC10	SE	IPI
FC11	SE	MF

**Table 4 sensors-25-06753-t004:** Classification accuracy of machine learning model of brake caliper (33EA).

Feature Combination	Machine Learning Model					
	SVM		KNN		DT	
	Normal	Defect	Normal	Defect	Normal	Defect
FC1 (SE-MAX)	100	99.6	98.5	99.6	100	99.6
FC2 (SE-RMS)	97.5	99.2	98.5	98.8	97.0	99.2
FC3 (SE-CF)	100	100	99.5	100	100	99.6
FC4 (SE-STD)	98.0	99.2	98.0	99.2	97.0	99.2
FC5 (SE-K)	100	100	98.5	99.6	100	99.6
FC6 (SE-FM0)	99.6	100	100	100	96.5	100
FC7 (SE-M6A)	99.6	100	99.6	100	100	99.6
FC8 (SE-M8A)	99.6	100	99.6	100	100	99.6
FC9 (SE-CLF)	100	100	100	100	100	100
FC10 (SE-IPI)	100	100	100	100	100	100
FC11 (SE-MF)	98.0	99.2	98.5	98.8	97.0	99.2

**Table 5 sensors-25-06753-t005:** Classification accuracy of machine learning models in the ablation study case 1.

Feature Combination	Machine Learning Model					
	SVM		KNN		DT	
	Normal	Defect	Normal	Defect	Normal	Defect
FC1 (SE-MAX)	76.5	70.4	74.5	71.7	75.5	67.5
FC2 (SE-RMS)	83.5	70.8	82.5	75.8	75.0	74.6
FC3 (SE-CF)	66.5	64.6	69.5	65.4	65.5	64.6
FC4 (SE-STD)	83.5	69.2	80.0	77.5	83.0	78.3
FC5 (SE-K)	74.5	60.8	72.0	62.5	74.5	60.4
FC6 (SE-FM0)	73.5	69.2	74.5	62.5	66.5	70.4
FC7 (SE-M6A)	67.0	52.1	70.5	59.2	69.5	61.3
FC8 (SE-M8A)	75.5	43.8	68.0	61.3	75.0	54.6
FC9 (SE-CLF)	67.5	68.8	71.0	64.6	67.5	70.0
FC10 (SE-IPI)	71.5	67.1	66.5	70.4	73.5	62.5
FC11 (SE-MF)	83.5	67.9	88.0	63.7	87.0	67.9

**Table 6 sensors-25-06753-t006:** Classification accuracy of machine learning models in the ablation study case 2.

Feature Combination	Machine Learning Model					
	SVM		KNN		DT	
	Normal	Defect	Normal	Defect	Normal	Defect
FC1 (SE-MAX)	97.0	97.5	97.0	98.8	95.0	97.9
FC2 (SE-RMS)	95.0	97.1	95.0	96.2	94.5	97.1
FC3 (SE-CF)	96.5	97.1	95.0	97.1	95.0	96.7
FC4 (SE-STD)	95.0	96.7	94.5	97.5	95.0	96.7
FC5 (SE-K)	95.5	97.1	95.5	96.7	95.0	96.2
FC6 (SE-FM0)	99.0	100	99.5	100	98.5	99.2
FC7 (SE-M6A)	97.1	95.0	95.0	97.1	94.5	97.1
FC8 (SE-M8A)	95.0	96.7	95.0	97.1	95.0	96.7
FC9 (SE-CLF)	99.0	99.6	99.0	99.6	99.5	98.8
FC10 (SE-IPI)	98.5	99.6	98.0	99.2	97.0	100
FC11 (SE-MF)	95.0	96.7	95.0	96.7	94.5	97.1

## Data Availability

The original contributions presented in this study are included in the article. Further inquiries can be directed to the corresponding author.

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
