# Peer review of "Real-Time Defect Identification in Automotive Brake Calipers Using PCA-Optimized Feature Extraction and Machine Learning"

_sensors, 2025, doi:10.3390/s25216753_

Round 1
Reviewer 1 Report
Comments and Suggestions for Authors
In this work is presented a non-contact automated impact-acoustic measurement system (AIAMS) for real-time detection of manufacturing defects in automotive brake calipers. The proposal is interesting but some issues need to be addressed.
- It should better if the manuscript is written in an impersonal way, please revise the whole manuscript and replace all sentences like "we propose" with "in this work is proposed"
- The introduction can be improved, the current version in very restrictive. The authors should highlight the most critical problems that are solved with this proposal.
- Section 3, methodology, can be also improved. First, it should be better if the overall flow chart of the method is presented at the beginning of this section; then , describe in a detailed way all performed steps.
- The description of results can be improved, if necessary, include a comparison with other similar works.
- Can the authors include ablation test?
- Are there some restrictions to be considered previous to the implementation of the method?
- What are planning the authors for future work? Include a brief description in the conclusion section.
Reviewer 2 Report
Comments and Suggestions for Authors
This paper studied a non-contact automated impact-acoustic measurement system for real-time detection of manufacturing defects in automotive brake calipers, a key component of the electric parking brake system. The article has a certain engineering application background. The following issues need to be noted:
- The applicability of the Principal Component Analysis method to this paper and the specific usage process need to be detailedly listed in Section 3.2, and certain comparative examples need to be provided to verify its applicability.
- Figure 2 is not clear. The analysis process of the proposed method needs to be elaborated in detail. The coordination between the image and the text should be particularly emphasized.
- This paper employs the method of machine learning, and the process of obtaining and processing the data needs to be presented.
- The discussion section of the fifth chapter needs to be further classified and presented in the form of charts and diagrams, so as to effectively express the characteristics of the obtained laws.
- What is the applicable scope of the detection and analysis methods proposed in this article? Do they meet the requirements for widespread application?
- It is necessary to add more examples of the existing methods and compare them with the existing ones to demonstrate the feasibility of the methods proposed in this paper.
Reviewer 3 Report
Comments and Suggestions for Authors
Comment
1.This research focuses on detecting only one type of micro-defect, which demonstrates the potential of the system well. However, the developed system has not been tested with a wide range of defect types. In the actual manufacturing process, other defects may occur, which represents one of the main limitations of this study. It is recommended to discuss further whether the proposed system can be applied to other defect types or to add preliminary experiments with 1–2 more defect types to increase the comprehensiveness of the study results.
2.The experiments in this paper were conducted with a single caliper model, which may have different shapes, materials, and surface textures on the measured acoustic signals. To increase coverage and applicability, it is proposed to further discuss whether the proposed method can be applied to calipers with different shapes or materials.
3.Although the use of a simulated conveyor system in this study is a good approach, in a real factory environment, noise, vibration, and temperature fluctuations are often much higher. Therefore, discussing how these factors may affect microphone accuracy or signal stability, and suggesting possible improvements for maintaining system performance under real conditions, would enhance the discussion section.
4.
In real-world manufacturing, the number of defective parts may be very small compared to normal parts, which is different from the dataset used in this study (5 normal and 6 micro defects). Therefore, how will the developed model perform under imbalanced data conditions?
In future work, techniques such as oversampling, cost-sensitive learning, or data augmentation could be considered to improve model stability. And if the authors can provide approximate data of actual production, such as the number of parts per lot and the average defect rate, This would also help readers better understand the industrial context of the study.
5.
The manuscript does not provide detailed specifications of the key measurement devices used in the experimental setup, particularly the impact hammer and microphone.
Since these instruments directly influence the reliability and repeatability of acoustic signal acquisition, the authors should include the following information:
- Type and model of the impact hammer (e.g., instrumented or automatic system)
- Impact force or range used during testing
- Microphone specifications (sensitivity, frequency range, sampling rate, and positioning distance)
- Any signal conditioning hardware such as preamplifiers or filters
Including this information would greatly enhance the reproducibility and transparency of the experiment, helping ensure that future researchers can accurately replicate the results.
Round 2
Reviewer 2 Report
Comments and Suggestions for Authors
There are no further technical issues. I agree to accept.
Reviewer 3 Report
Comments and Suggestions for Authors
To the Authors,
Thank you for your thorough and systematic revisions to the manuscript. I am pleased to see that all the concerns raised in my initial review have been satisfactorily addressed. The manuscript is now significantly stronger, clearer, and more transparent. The revisions have successfully addressed all my comments.